# Effect of Drought Stress on Degradation and Remodeling of Membrane Lipids in *Nostoc flagelliforme*

**DOI:** 10.3390/foods11121798

**Published:** 2022-06-18

**Authors:** Meng Wang, Qiang Zhu, Xiaoxu Li, Jinhong Hu, Fan Song, Wangli Liang, Xiaorong Ma, Lingxia Wang, Wenyu Liang

**Affiliations:** School of Life Sciences, Ningxia University, Yinchuan 750021, China; wm123r@163.com (M.W.); qzhu2008@163.com (Q.Z.); lixiaoxu_2020@163.com (X.L.); hjh199633@163.com (J.H.); sf_0726@163.com (F.S.); lwl2090197695@163.com (W.L.); mxr19961222@163.com (X.M.); wang_lx0218@163.com (L.W.)

**Keywords:** *Nostoc flagelliforme*, lipid metabolism, drought stress, OCFAs

## Abstract

*Nostoc flagelliforme* is a kind of terrestrial edible cyanobacteria with important ecological and economic value which has developed special mechanisms to adapt to drought conditions. However, the specific mechanism of lipidome changes in drought tolerance of *N. flagelliforme* has not been well understood. In this study, the ultra-high-performance liquid chromatography and mass spectrometry were employed to analyze the lipidome changes of *N. flagelliforme* under dehydration. A total of 853 lipid molecules were identified, of which 171 were significantly different from that of the control group. The digalactosyldiacylglycerol/monogalactosyldiacylglycerol (DGDG/MGDG) ratio was increased. The amount of wax ester (WE) was sharply decreased during drought stress, while Co (Q10) was accumulated. The levels of odd chain fatty acids (OCFAs) were increased under dehydration, positively responding to drought stress according to the energy metabolism state. In conclusion, the lipidomic data corroborated that oxidation, degradation, and biosynthesis of membrane lipids took place during lipid metabolism, which can respond to drought stress through the transformation of energy and substances. Besides, we constructed a lipid metabolic model demonstrating the regulatory mechanism of drought stress in *N. flagelliforme*. The present study provides insight into the defense strategies of cyanobacteria in lipid metabolic pathways.

## 1. Introduction

Drought is a major abiotic stress and causes a huge loss in crop yields worldwide, affecting around 64% of the world’s land area [1]. Water deficiency results in the overproduction of ROS and decrease in photosynthesis, leading to growth impairment [2]. However, plants, including lower plants, have developed numerous strategies, which include morphological, physiological, and molecular levels, for coping with adverse effects of drought stress [3,4]. Cyanobacteria are ancient prokaryotes performing oxygenic photosynthesis with wide ecological distribution. Therefore, cyanobacteria have to cope with various abiotic stresses, such as extreme dryness and irradiation, in terrestrial ecosystems. For example, increased EPS contents, inhibition of photosynthesis, or enhanced molecular chaperone expression are crucial for cyanobacteria acclimation to drought stress [5,6,7]. Cyanobacteria have developed various mechanisms to adapt to drought stress, but the key regulatory factors related to drought tolerance are different among cyanobacterial species.

*Nostoc flagelliforme* is a terrestrial nitrogen-fixing cyanobacterium species, which is distributed in arid and semi-arid steppes in the west and northwest of China [8]. It is wrapped in a gelatinous sheath and exists in the form of a filamentous colony. As a pioneer species, *N. flagelliforme* is capable of fixing large amounts of atmospheric nitrogen to nourish barren land, and thus has great potential for application in environmentally safe and sustainable agriculture [9]. In addition, this cyanobacterial species is edible and has great medicinal and economic values [10,11]. Interestingly, the Chinese have regarded it as a food delicacy for about 2000 years [12]. A study reported that *N. flagelliforme* contains eight amino acids which are essential for human health [13].

The colonies of *N. flagelliforme* show strong tolerance to severe stresses and survive for several decades under extremely arid conditions and rapidly resume physiological and metabolic activities after reabsorbing water [14]. Therefore, *N. flagelliforme* has emerged as an ideal material for explaining stress adaptation in the xeric environment [15]. Previous studies mainly focused on its structure and physiological characteristics [16,17]. To date, genome, transcriptome, and protein succinylation studies have revealed that energy metabolism, scavenging of reactive oxygen species, and photosynthesis were effective strategies for *N. flagelliforme* in response to desiccation stress [14,18,19]. In addition, metabonomic analyses have revealed the dynamic changes in the metabolite responses of *N. flagelliforme* in response to rehydration and dehydration processes [10]. Thus, the overlap or combination of morphological, physiological, and molecular response is its defense strategy against stress [15]. Although some studies have been performed to reveal the drought tolerance mechanism of *N. flagelliforme*, a lack of lipid profile data has constrained extensive and intensive research on this cyanobacteria species.

Lipids are important components of biological systems and are associated with many crucial biological roles, such as being the organizing component of plasma membranes, energy generation and storage, signaling functions in the metabolic regulation, and participation in cell–cell interactions as ligands or mediators [20]. Phospholipid compounds such as PE and PC are important for maintaining membrane fixation in plants. An excellent example is that the PC and PE lipid species had higher accumulation levels in 4/10 genotypes from the *Lolium-Festuca* complex, which improved the stability of the extra-chloroplast membrane and had a faster response to drought conditions [21]. Besides, phospholipids also serve as signaling molecules for information transduction by activating specific protein phosphatases and kinases, thereby activating downstream signal pathways and leading to physiological reactions [22]. PA has been identified as an important lipid second messenger, and the content of PA increased significantly in tall fescue during drought stress, assisting in developing early defense mechanisms [23]. Overall, lipids possess two major roles in plant adaptation to drought conditions. On one hand, they act as signal mediators; on the other hand, they play an important role in mitigating the detrimental effects of stress [24].

In order to better understand the lipid metabolism of *N. flagelliforme* during the dehydration, to clarify the corresponding pattern, and further reveal the underlying molecular mechanism as well as related metabolic pathways, we investigated the lipid profile of *N. flagelliforme* samples obtained after different drought stress treatments using the untargeted lipidomics approaches for identifying the lipid molecular species. Conclusively, the present work supports the view that the lipid composition and lipid metabolism were significantly changed in different drought stress. The acquaintance of these differences will expand our knowledge of the drought tolerance in *N. flagelliforme* and help to analyze the adaptation of cyanobacteria to drought stress to match lipid production strategies, and also help to provide a theoretical basis in the field of lipid metabolism for the effective use of nitrogen-fixing cyanobacteria in agriculture.

## 2. Results

### 2.1. Generation of Lipid Profiles in N. flagelliforme under Drought Stress

The lipid composition of *N. flagelliforme* under drought stress was analyzed using an UHPLC-MS. A total of 853 lipid species were identified and quantified (Appendix A), which represented six main categories including GL, GP, SP, SL, FA, and PL. The identified lipids were further classified into 37 different lipid classes, of which the most common lipids comprised 168 TG, 100 DGDG, 80 DG, 75 MGDG, 35 SQDG, 29 PC, 28 PE, and 25 PG (Appendix A).

Principal component analysis (PCA) was performed on the dataset, which showed that there was a clear distinction between the control and drought stress groups (Figure 1). It was worth mentioning that PC1 and PC2 accounted for 30.46% and 17.48% of the overall variance, respectively, suggesting that dehydration induced change the lipid profile of *N. flagelliform*. The OPLS-DA score plot showed a clear trend of separation between groups, and the model showed high R^2^ and Q^2^ statistics (Appendix A), indicating good fitness and high predictability. The results showed that the reliabilities of the obtained lipidome data set was relatively high and could be used for subsequent analysis.

To further confirm the contribution of each lipid molecule species to the group separation and to search for potential lipid markers of the *N. flagelliforme* under drought stress, the VIP value was used to explore the variables that have a greater impact on the OPLS-DA model. In fact, most higher VIP values had greater influence on variables. The significantly different lipids were selected using the VIP > 1 and combined with Student’s *t*-test (*p* < 0.05) as key biomarkers during drought stress. Appendix A show the significant differences in lipids identified in the treated samples compared to the control group under drought stress. The top five lipids with the highest VIP values were DG (29:1), SQDG (17:5/23:2), SQMG (16:0), MGMG (16:0), and MGDG (16:1/18:3). These results indicated that the five lipid molecule species underwent significant changes during drought stress and can be used as biomarkers of drought stress in *N. flagelliforme*.

### 2.2. Comparative Analysis of DELs under Drought Stress

The changes of lipid molecule species in *N. flagelliforme* under drought stress are provided in Figure 2. A total of 171 DELs were identified from the three pairwise comparisons: LB vs. LA, LC vs. LA, and LD vs. LA. In comparison to the LA group, 64 (31 up-regulated and 33 down-regulated), 94 (69 up-regulated and 25 down-regulated), and 98 (75 up-regulated and 23 down-regulated) DELs were detected in the LB, LC, and LD groups, respectively (Figure 2 and Appendix A). Interestingly, the up-regulation of DELs was 3.26 times higher than the down-regulation of DELs in the LD group, and the up-regulation of DELs was 0.94 times lower than the down-regulation of DELs in LB group. These changes implied that the biosynthesis of lipid molecule species mainly occurred during the drought stress process, whereas the degradation may have occurred during rehydration. Comparative analysis revealed that the number of overlapping samples between the LC and LD groups were higher than those in the other two comparisons (LB vs. LC and LB vs. LD). A total of nine DELs were conserved and shared by all three pairwise comparisons. These lipid molecule species included DG (18:1/18:3), MGDG (16:0/18:1), MGDG (16:1/18:3), SQDG (17:5/23:2), Cer (30:0), CerP (30:2), PC (16:1/18:1), PC (18:1/18:1), and Co (Q10). The results suggest that they may play an important role in improving the drought resistance of *N. flagelliforme.*

The composition of lipid classes was changed in different water loss groups under drought stress (Figure 3). The proportions of GL to the total number of lipid species was 40% in the LB group compared with those in the LA group (Figure 3A). Among them, the proportion of DG and TG was 20%, respectively. The percentage of TG and DG decreased to 7% and 11% in the LC group compared with LB. However, the number of MGDG species increased, accounting for 15% of all DELs (Figure 3B). These results indicate that with the aggravation of drought stress, the number of different glycerolipids decreased and was accompanied by the increase in the number of saccharolipids. During drought stress, phospholipid molecules decreased and lysophospholipids accumulated in the LC vs. LA group. In the LD vs. LA group, saccharolipids accounted for the largest proportion among the major lipid types. In addition, the total carbon number composition of DELs in *N. flagelliforme* was different under drought stress (Appendix A). In comparison with the LA group, the total carbon number of DELs ranged from C21 to C62 in the LB group. Among them, the C36 species, which contained 11 lipid species (mainly TG and DG), were particularly prominent in the LB group (Appendix A). In the LC vs. LA comparison, the length of detected total carbon atoms varied from 19 to 64 carbons in the LC group. C43 and C44 were rich in lipid molecule species, among which the fatty acid chain composition was mainly (16:1) and (18:1) (Appendix A and Appendix A). The carbon atom number of DELs ranged from C20 to C66 in LD the group compared with the LA group. C44 species was composed of 11 lipid species, whereas C43 species was composed of 10 lipid species (Appendix A). Their fatty acid chains were mainly composed of (16:0), (16:1), and (18:1). These results showed that the number of carbon atoms of lipid molecules was increased with the dehydration process and lipid metabolism was dominated by biosynthesis.

### 2.3. Quantitative Changes of Lipid Molecular Species in N. flagelliforme under Drought Stress

The changes in glycerolipid content and fatty acid compositions, including TG and DG, are summarized in Figure 4 and Appendix A. In comparison with the control group, DG class showed a trend of first increasing and then decreasing, and the total lipid content of DG class was the highest in LB the group, with an increase of 47.2%. Similarly, TG class showed the same trend and was up-regulated in the LB and LC groups. The (C29:1)- and (C31:4)-DG species were dominant in the DG class and were the major contributors to the increased DG content. Compared with the LA group, the lipid molecule species, including (16:0, 16:0), (16:0, 16:1), (18:3, 18:2), and (18:3, 18:3), were significantly up-regulated in the LD group. The (33:3)- and (12:0, 10:3, 11:3)-TG were the main components that determined the increase or decrease of TG. The relative intensities of (18:0, 18:0, 18:2)- and (18:0, 18:1, 18:3)-TG species were higher in the four groups, which were conducive to the accumulation of total TG.

Differences in the quantitative intensity of glycerolphospholipids between the control group and stress treatment groups are shown in Figure 5. PC was the most abundant phospholipid, and the contents of PC, PG, PI, and PS were all increased in the LD group (*p* < 0.001), whereas the content of PE was reduced in the LB and LC groups (*p* < 0.05). PG, PI, and PS, which constituted phospholipids, contributed approximately 14.40%, 11.55%, and 3.87% to the total glycerolphospholipids, respectively. The content of (16:1, 18:1)-, (18:1, 18:1)-, (19:1, 16:0)-, and (35:2)-PC species was drastically higher in the LD group (*p* < 0.001), which was the major contributor for the increased amount of PC (Appendix A). The decrease in PE of the LB and LC groups was mainly due to the down-regulation of its component lipid molecules (16:1, 18:1)-, (34:2)-, and (36:2)-PE. Lysophospholipids, such as LPC, LPE, and LPG, were accumulated under drought stress. The accumulation of LPC, LPE, and PLG in the LD group was 6.68, 15.26, and 1.48 times higher than that in the control group, respectively. Lysophospholipids were composed of a few molecular species, including (16:0), (16:1), and (18:1) carbon molecules. Both LPC and LPE were composed of (16:1) and (18:1) species and had an upward trend. Interestingly, the amount of 18:1-LPG was found obviously higher in the LC group.

Saccharolipids, including DGDG, MGDG, and SQDG as main structural lipids, are involved in the composition of photosynthetic membranes. In fact, the content of DGDG and MGDG decreased slightly in the LB group and then increased in the LC and LD groups (*p* < 0.001) (Figure 6). The (16:1,16:1)- and (18:3,18:3)-DGDG species were dominant in the DGDG class and were the major contributors to the increased DG content. In addition, the (16:1,18:3)- and (18:3,18:3)-MGDG species were dominant in MGDG (Appendix A). Furthermore, the accumulation of DGMG and MGMG increased in the later stage of stress, and MGMG in particular increased most at 75% water loss of the sample. The content of SQDG gradually increased and the accumulation reached the maximum level in the LD group during dehydration. DGMG, MGMG, and SQMG fatty acyl chains were composed of a small number of lipid molecules, mainly (16:0), (16:1), (18:1), and (18:3). The acyl chains of (16:1, 18:2) carbon molecules were coincidently dominant in DGDG, MGDG, and SQDG classes and were simultaneously enhanced in the LD group (*p* < 0.001).

Sphingolipids, including Cer, CerP, GM3, and So, were the main structural lipids found in *N. flagelliforme* under drought conditions (Figure 7). In response to drought stress, the change trends of Cer and So were similar; the change in Cer and So intensity was not obvious at the initial stage of stress (water loss 30%), but then they were down-regulated (*p* < 0.001). The major contributors to decreased Cer were (18:0, 14:0) and (34:0) species (Appendix A). Besides, the content of So with the (20:0) acyl chain was drastically lower in the LC and LD groups, which was the major reason for the decreased So. However, CerP and GM3 had exactly the same trend where they both showed a gradual increase with drought developed, and reached the maximum value at 100% water loss. CerP class consisted of only one fatty acyl chain, i.e., a (32:0) carbon molecule, while the (30:0) and (30:2) species were dominant in GM3. The transfer of acyl chains from (18:0, 14:0)-, (28:0)-, (30:0)-, (34:0)-, (34:1)-, (36:0)-, (38:0)-, (38:1)-Cer, and (20:0)-So to (30:2)-CerP and (30:1)- and (30:2)-GM3 was the main reason for reduced Cer and So and increased CerP and GM3 contents.

WE consists of only one fatty acyl chain, which is a (4:0, 16:0) carbon molecule. Compared with the control group, the expression intensity of WE in the LC and LD groups was reduced by 48% and 51%, respectively (Figure 8). However, the amount of Co (Q10) was increased with aggravation of drought stress in *N. flagelliforme* (*p* < 0.001), and the maximum rate was observed in the LD group, which was 3.54 times higher than the initial value.

### 2.4. Determination of Physiological Indexes in N. flagelliforme under Drought Stress

For the purpose of understanding the changes in lipid metabolism, eight enzymes related to lipid metabolism were selected, more specifically, those involved in lipid oxidation, degradation, and DGDG biosynthetic pathways (Figure 9). The biosynthesis of DGDG is catalyzed by a series of enzymes, including GPAT, AGPAT, PAP, MGD1, and DGD1. Here, GPAT catalyzes the first reaction in the biosynthesis of DGDG to form LPA. Compared with the control group, the GPAT enzyme activity increased and was highest in the LB group (Figure 9A). AGPAT is involved in the synthesis of PA, and then dephosphorylation of PA produces DG by PAP. The activity of AGPAT began to increase in the late stage of stress (75% water loss), while the activity of PAP increased continuously with the dehydration process (Figure 9B,C). Type A MGDG synthase 1 (MGD1) catalyzes the formation of MGDG from DG, which is then catalyzed by DGD1 to produce DGDG. The activity of DGD1 in the LD group was 14.37% higher than that in the control group. (Figure 9E). The products catalyzed by PLD and PLC participate in lipid peroxidation. PLD and PLC participate in catalyzing PC conversion into PA and DG. There was no difference in PLD activity at the statistical level, while the activity of PLC enzyme exhibited a trend of first increasing and then decreasing (Figure 9F,G). PLA2 is involved in the degradation of structural lipids, such as PC and PE. The PLA2 enzyme activity in the LD group was significantly increased, which was 1.03 times of that in the control group (Figure 9H). ACOX1 is a rate-limiting enzyme that participates in fatty acid β-oxidation of peroxisome and is involved in the dehydrogenation of acyl-CoA. The enzyme activity of ACOX1 was changed, but the change was not obvious (Figure 9I). However, the ACOX1 content was accumulated with the stress aggravation and reached the maximum in the LD group (Appendix A), indicating that lipids were oxidized in *N. flagelliforme* during severe dehydration, and energy is accumulated to resist drought. The above results indicated that the enzymes catalyzing lipid metabolism underwent significant changes under drought stress and may directly lead to changes in the composition and structure of membrane lipids.

The net photosynthesis increased at the beginning at 0% water loss, reached the maximum rate at about 30% water loss, decreased further with an increase in dehydration, and finally decreased to the lowest rate at 100% water loss (Figure 9I). The maximum net photosynthetic rate increased by 1.47 times compared to that of the initial value. Similarly, ΦPSII reached the maximum value at 30% water loss and decreased with an increase in drought stress (Figure 9J). Compared with the control group, ΦPSII decreased when water loss of algae was about more than 75% and reached almost zero at 100% water loss. These results suggested that water deficiency leads to a decrease in the photosynthetic capacity of the colony, which may be due to the changes in the composition and structure of the photosynthetic membrane.

As shown in Figure 9K, the change in the MDA content in the early stage of drought stress was not obvious different. However, the MDA content was increased with the aggravation of drought stress (*p* < 0.05). When the water loss was 100%, the content of MDA increased by 166.73% compared with that of the control group. The results showed that the membrane lipids were subjected to a certain degree of oxidation reaction under drought conditions, thus affecting the degradation and synthesis of lipids.

## 3. Discussion

Environmental stress, especially water stress, is a global problem that restricts plant growth and development. Being terrestrial cyanobacteria, *N. flagelliforme* has adopted various strategies to deal with drought stress and has emerged as an ideal material to explain stress adaptation in drought environment [19,25]. In the present study, we provided comprehensive insight into the composition of membrane lipids of different dehydrated *N. flagelliforme* samples using lipidomic approaches. To the best of our knowledge, this is the first study to completely characterize the lipidomic profile of *N. flagelliforme* under drought conditions.

### 3.1. Changes in Membrane Lipids in N. flagelliforme Are Associated with Photosynthesis under Drought Stress

Cyanobacteria are prokaryotes with oxygenic photosynthesis and have become model organisms for studying photosynthetic mechanisms [26]. The thylakoid membranes, mainly composed of MGDG, DGDG, SQDG, and PG, are photosynthetic reaction sites in cyanobacteria [27]. As structural lipids, DGDG and MGDG constitute the backbone of the plastid membrane, which is necessary for photosynthesis [28]. It is worth noting that the cone-shaped MGDG could not be arranged in the lamellar structure but can be transformed into cylindrical DGDG to maintain the stability of the thylakoid membrane [29,30]. A recent study has shown that variation trends of galactolipids differed among molecular species during water stress in winter wheat seedlings. Importantly, (36:6)- MGDG and (36:6)- DGDG mainly play a role in stabilizing the membrane system, whereas (34:3)- MGDG and (34:3)- DGDG are involved in the lipid bilayer of photosynthetic membranes and photosynthetic protein–cofactor complex [31]. In addition, the reduction in plant photosynthesis observed under stress is a direct result of insufficient DGDG [32]. Interestingly, it was found that the lack of water in cowpea leaves promoted the biosynthesis of DGDG and increased the tolerance of plants to arid environments [33]. In the present study, the enzymes in the metabolic pathways of UDP-Gal-dependent DGDG synthesis, such as GPAT, AGPAT, PAP, MGD1, and DGD1, were all significantly up-regulated in *N. flagelliforme* (Figure 9), thereby increasing the content of MGDG and DGDG. The findings of the present study showed that the stress-induced DGDG synthesis can cause membrane lipid remodeling, which had great significance in regulating membrane lipid composition, fatty acyl unsaturation, and membrane fluidity.

Another feature of plant tolerance to stress is an increase in the ratio of DGDG to MGDG [34]. The DGDG/MGDG ratio was increased during the cold acclimation of *Lepidium sativum* (cress), which may enhance the membrane interaction [35]. In addition, the DGDG/MGDG ratio of *Thymus* under water and drought conditions was 0.93 and 0.97, respectively [24]. In this study, the ratio of DGDG to MGDG increased from 0.19 to 0.26 during drought stress in *N. flagelliforme* (Figure 6). Indeed, the increase in DGDG/MGDG ratio is due to the increase in (C16:1, C18:3)- and (C18:3, C18:3)-DGDG content and decrease in (C47:12)-MGDG content. In addition, unsaturated fatty acid content determines the fluidity of the lipid membrane, and the variable content of unsaturated fatty acids can improve the tolerance of plants to environmental stress. Isoprene can effectively maintain the integrity of the thylakoid membrane by increasing the level of unsaturated fatty acids [36]. In the present study, the levels of lipid molecular species (C47:14)-, (C48:14)-, and (C50:16)-DGDG were increased significantly, which extremely improved the lipid unsaturation in *N. flagelliforme* (Figure 6). Maintaining a high level of polyunsaturated fatty acids in thylakoid lipids helps in maintaining efficient photosynthesis.

In cyanobacteria, the thylakoid membrane is the site of photosynthesis, and SQDG is one of the main components of the photosynthetic membrane. Recently, untargeted lipidomic analysis of *Synechocystis* sp. PCC 6803 revealed significant differences in lipid content, and the most prominent difference in the relative levels of high-abundant lipid species was observed in the SQDG lipid class [27]. Furthermore, it was found that SQDG has a specific function of maintaining PSII properties in *Synechocystis* sp. PCC 6803, and PSII has a species-specific demand for SQDG [37]. SQDG participates in the structural composition of the PSII complex and interacts with the PSI complex to maintain the structure of the PSI trimers in *Thermosynechococcus elongatus* [38]. Therefore, the significant up-regulation of SQDG level detected in *N. flagelliforme* may be part of a wider lipidomic adaptation program (Figure 6). PG is a component of the thylakoid membrane, and the relative expression intensity shows that PG has the same changing trend as that of DGDG and MGDG, indicating that PG plays an important role in photosynthesis. In fact, PG has been shown to be involved in the activities of photosystems I and II [39].

The reduction in photosynthetic activities is one of the effective strategies of *N. flagelliforme* for managing significant water loss [40]. Previous studies have shown that the net photosynthetic rate (Pn) in *N. flagelliforme* reached a maximum at 30% water loss and then decreased with any further dehydration [41]. It is worth noting that the findings of the present research are consistent with the above reports, in which the net photosynthesis was initially increased and then decreased. Another interesting phenomenon is that the changing trend of effective quantum yield of PSII under drought stress is similar to that of net photosynthesis (Figure 9). The result showed that photosynthesis of *N. flagelliforme* decreased under water stress. Although the photosynthetic activity of *N. flagelliforme* was decreased under drought stress, it can be fully restored within a few hours to several days after rehydration [41]. In drought-tolerant plants, maintaining the structural integrity of photosynthetic organs during dehydration is essential for the recovery of photosynthetic activity after rehydration [7]. In this study, the membrane structure of thylakoids was intact under drought stress (Appendix A), and there was no apparent change between control and drought stress materials. Similarly, the structure of the thylakoid membrane between the dehydrated and rehydrated colonies did not exhibit much difference, and the photosynthetic activity was almost zero after 48 h of drought, but the photosynthetic activity significantly increased after 4 h of rehydration [16]. Thus, membrane lipid remodeling and the increase in unsaturated fatty acids might be the main reasons for maintaining its integrity, which is beneficial for the recovery of photosynthetic activity after rehydration.

### 3.2. Membrane Lipid Peroxidation Was Alleviated by Lipid Allocation and Accumulation

ROS are highly reactive and initiate lipid peroxidation, and thus, excessive ROS might induce loss of cell membrane structural integrity [42]. It has been reported that ROS are dramatically increased in response to drought stress, resulting in oxidative damage to cellular components [43]. MDA is an important parameter indicating the degree of membrane damage caused by ROS [44]. Moreover, the accumulation of lipid second messenger PA can further promote the production of free radicals and H_2_O_2_, thereby causing serious damage to membrane lipids [29]. A study found that the content of H_2_O_2_ increased significantly in *N. flagelliforme* under different drought stress conditions, but was lower than that in the control group [45]. In our work, interestingly, we found that the MDA content in *N. flagelliforme* was increased under drought stress (Figure 9). However, PA level was not significantly different in lipid molecule species in *N. flagelliforme* under stress (the relative expression intensity of PA is shown in Appendix A). In general, there are two pathways to generate PA: the direct product of PLD and the secondary product of the PLC pathway [46]. The findings of the present study showed a significant change in PLC activity rather than PLD activity (Figure 9), suggesting that PA production is primarily via the PLC pathway. The activity of PLC decreased at the later stage of stress, which reduced the accumulation of PA. In addition, the possible reason for these results is that PA is the precursor of all galactolipids, TG, and phosphoglycerolipids, the latter of which contributes more to drought stress tolerance [47]. Reasonable transformation of PA is of great significance to lipid metabolism and plasma membrane structure.

It is worth noting that Co (Q10), as a redox active molecule, is embedded in the hydrophobic domain of phospholipid bilayer of the cytoplasmic membrane and plays a crucial role in scavenging harmful ROS in the process of disulfide bond formation [48]. The relative expression intensity of Co (Q10) in *N. flagelliforme* was significantly accumulated with the increase in dehydration (Figure 8). It is worth mentioning that antioxidant lipids such as Co (Q10) were also significantly accumulated to alleviate the lipid peroxidation of *N. flagelliforme* during the process of plasma membrane oxidation by active substances.

### 3.3. Response of Membrane Lipid Degradation to Drought Tolerance in N. flagelliforme

In general, lipids are considered as membrane structures and energy storage molecules, however, recently they have been recognized as powerful signaling molecules that regulate various cellular responses. Lysophospholipids (LPs) are present in tissues in low concentrations under non-stress conditions, however, as the membrane lipids are hydrolyzed, they accumulate and release into extracellular spaces as signaling lipids to initiate signaling pathways [49]. Previous studies reported that a significant increase in LPC and LPE levels is a response to stress [29]. LPs containing a single fatty acid chain are produced by the hydrolysis of PLA2 [50]. In the present study, the levels of PC and PE decreased to a certain extent during the stress, and the activity of PLA2 was up-regulated under stress. Thus, LPC, LPE, and LPG were accumulated in different quantities (Figure 5). The results showed that the significant increase in LPs might be related to the activation of signaling pathways and act as signaling lipids to initiate stress response. Moreover, PLA2 hydrolyzes complex lipids to produce free fatty acids and activate the fatty acid degradation pathway [47]. The degradation of fatty acids can generate enough energy and more polyunsaturated fatty acids to maintain the fluidity of cell membranes in response to stress [47]. Fatty acid β-oxidation is the main form of fatty acid degradation to produce energy and is also the central process in the α-linolenic acid metabolic pathway [47]. In this study, the activity of ACOX1 did not distinctly change, but its content was significantly increased (Appendix A). Moreover, ACOX1 activated the β-oxidation pathway of fatty acids under drought stress, and linolenic acid was accumulated to a high level in *N. flagelliforme*. In summary, lipid degradation may activate related stress signaling pathways and provide the required energy for *N. flagelliforme*.

During the evolution from aquatic to terrestrial environment, plants have obtained various protective functions against environmental stress, and thus, one of the important adaptation characteristics is the formation of the protective layer. The surface of grapevine berry is covered with a cuticle composed of cutin and various lipophilic wax compounds, which help protect the fruit from environmental factors such as pests, mechanical shock, or radiation [51]. In our study, when the stress was aggravated in *N. flagelliforme*, the lipid molecular composition (C4:0, 16:0) of WE significantly decreased, which accelerated lipid degradation caused by drought (Figure 8). However, Wan et al. [52] found that the wax content decreased and the number of plastid lipids increased (such as 36:6-DGDG) during the development of “Newhall” navel orange. This complementary carbon flux could promote a more effective adaptation of the plant to the environment. The reason for this conversion is that fatty acids can be used for both the biosynthesis of various aliphatic wax components and the biosynthesis of plastid-related membranes [53]. However, the specific mechanism of fatty acid and plant wax biosynthesis is still unclear and needs further research. The findings of the present study showed that the content of various plastid lipids such as DGDG and SQDG was increased under drought stress (Figure 6). Therefore, the decrease in WE in the colony was due to the fatty acid substrates competing for the synthesis of plastid lipids for maintaining membrane stability.

### 3.4. Changes in Odd-Chain Fatty Acids (OCFAs) in Response to Drought Stress

Fatty acids naturally occur in prokaryotic and eukaryotic organisms. In recent years, OCFAs have attracted increasing attention due to their unique pharmacological effects and applications in agriculture and industry [54]. It has been reported that a low amount of OCFA was identified in cyanobacterium *Synechococcus* sp. PCC 7002 [55]. It is worth noting that propionyl-CoA can be converted into succinyl-CoA under some stress conditions, and then generate ATP in the tricarboxylic acid cycle, however, OCFA may be generated as a by-product, and its large accumulation is conducive to more energy generation [55]. In addition, fast-dividing PCC 6803 cells contain lower lipid levels of OCFA than that in the slower cells, which directly affects the availability of resources and precursors [27]. In this study, several OCFA lipid molecule species, such as (C29: 1)-DG, (C47: 14)-DGDG, (C35: 2)-MGDG, and (C45: 4)-PS, accumulated in *N. flagelliforme* during drought stress. Therefore, we propose that the level change in OCFAs is an important physiological characteristic that is adaptive to the energy state in *N. flagelliforme* under drought stress.

The possible mechanism of lipid metabolism in *N. flagelliforme* under drought stress was proposed based on the data obtained (Figure 10). Drought stress causes changes in the physiological levels of *N. flagelliforme*, which regulates lipid metabolism pathways through enzyme activity and alters the lipid composition.

## 4. Experimental

### 4.1. Materials

Colonies of *N. flagelliforme* were collected from the eastern region of Helan Mountain (38.41′ N, 105.95′ E) in Ningxia, China. The air-dried filamentous were washed with deionized water. Next, surface-sterilizing was performed by 75% ethyl alcohol for 30 s and washed with sterilized water three times. Most or all of the epiphytic microorganisms on the surface of *N. flagelliforme* can be removed by ethanol sterilization method [10]. The samples were divided into plastic nets as algal mats with a diameter about of 10 cm and then rehydrated in BG11 medium [12] (containing 1.5 g·L^−1^ NaNO_3_) for 1.5 h to ensure sufficient water absorption. After that, the samples were transferred to an incubator for restore physiological activity at 25 °C, 30% relative humidity, and continuous illumination of 40 μmol·m^−2^·s^−1^, and sprayed with BG11 medium every half hour for 4 h to absorb water. These colonies were pretreated for 1 h under the conditions of 25 °C, 30% relative humidity, and 400 μmol·m^−2^·s^−1^ light intensity, and kept under spray medium to maintain moisture. Water was removed from the surface of colonies and plastic nets, and the samples were collected immediately as sample LA (water loss 0%, control). For drought stress, the dehydration process was carried out continuously with 400 μmol·m^−2^·s^−1^ light intensity at 25 °C and 30% relative humidity. Water loss (WL, %) = (Ww − Wt)/(Ww −Wd) × 100%, where Ww is the initial wet weight, Wt is the instantaneous weight of samples measured at certain intervals, and Wd is the dry weight [16,41]. The samples LB, LC, and LD were dehydrated by 30%, 75%, and 100%, respectively. After stress treatment, the samples were immediately immersed in liquid N_2_ and then stored at −80 °C until further analyses.

### 4.2. Preparation of Lipid Samples

Lipids were extracted from eight independent biological replicates of *N. flagelliforme* at different drought stress using the methyl tert-butyl ether (MTBE) method [56]. Briefly, the 30 mg sample powder was homogenized with 200 µL of water and 240 µL of methanol. Then, 800 µL of MTBE was added for further extraction. The lipid phase was obtained by centrifugation at 14,000× *g*, 10 °C, for 15 min and then dried under nitrogen.

### 4.3. Lipid Analysis by UHPLC-Q-Exactive Orbitrap Mass Spectrometry

Lipids were chromatographically separated by ultra-high-performance liquid chromatography (UHPLC) Nexera LC-30A system using ACQUITY UPLC CSH C18 column (1.7 µm, 2.1 mm × 100 mm, Waters) and kept at a temperature of 45 °C. The injection volume was 3 µL. The elution gradient was carried out using the method reported by Liu et al. [29]. The flow rate was 300 μL·min^−1^. The mobile phase consisted of component A: 10 mM ammonium formate in acetonitrile and water (acetonitrile:water = 6:4, *v*/*v*) and component B: 10 mM ammonium formate in acetonitrile and isopropanol (acetonitrile: isopropanol = 1:9, *v*/*v*). The gradient elution procedure was performed as follows: 0–2 min, 30% B; 2−25 min, 100% B; 25−35 min, 30% B. The samples were placed in autosampler sampler t at 10 °C. Mass spectra were acquired by Q-Exactive Plus Orbitrap mass spectrometer (Thermo Fisher Scientific^TM^, Waltham, MA, USA) in simultaneous positive and negative mode. Electrospray ionization (ESI) analysis was conducted according to parameters reported by Zheng et al. [57]. The spray voltage, S-Lens RF Level, and MS1 scan ranges were set at 3.0 KV, 50%, and 200–1800, respectively, in positive mode and 2.5 KV, 60%, and 250–1800, respectively, in negative mode. The quality control (QC) was obtained by mix equal volume of each sample to determine the equilibrium chromatography-mass spectrometry system before injection, and to evaluate the repeatability of the system during the whole experiment.

### 4.4. Data Processing and Statistics

The raw MS data acquired from LC-MS were processed using Analyst (version 1.42) by extracting ions from the total ion chromatogram (TIC) [58]. The chromatographic peak areas were integrated by extracted fragment ions and aligned across the sample set to identify lipid molecules. The lipid molecular species were identified and quantified using LipidSearch software (version 4.1) (Thermo Fisher Scientific™, Waltham, MA, USA) on the basis of accurate mass and fragment matching. The main processing parameters were set as previously described [29]. For the data extracted from LipidSearch, lipids with RSD greater than 30% or missing values greater than 50% were removed. The abundance values were obtained using total peak area normalization using the Perato scaling method. The multivariate analysis was performed as reported earlier [20]. The Student’s *t*-test (*p* < 0.05) and VIP value were carried out for screening statistical significance of each group in differently expressed lipids (DELs), where VIP > 1 and *p* < 0.05 [29].

### 4.5. Detection of the Physiological Parameter

The activity of PLA2, PLC, PLD, GPAT, AGPAT, PAP, MGD1, and DGD1 were determined using an assay kit (Jingmei Biotechnology Co., Ltd., Yancheng, China) and strictly in accordance with the manufacturer’s instructions. The measurement principle is as follows: the purified enzyme antibody was coated with microtiter plate wells, then add enzyme to the wells, combined antibody with HRP labeled, become antibody-antigen-enzyme-antibody complex. TMB substrate solution was added, which turned blue under HRP enzyme catalysis, and sulfuric acid solution was added to terminate the reaction. The absorbance value was measured by spectrophotometry at 450 nm. The enzyme activity was determined by taking 1 g sample and added with PBS solution (pH 7.3 ± 0.1) for full homogenization. The solution was centrifuged at 855× *g* for 20 min and the supernatant was carefully collected for subsequent determination. The measurement of the content of MDA, a lipid peroxidation product in *N. flagelliforme*, was performed in strict accordance with the instructions of the MDA content kit of Suzhou Comin Biotechnology Co., Ltd. (Suzhou, China). The determination principle is as follows: MDA is condensed with TBA and the product has a maximum absorption peak at 532 nm. The content of lipid peroxide in the sample could be estimated after colorimetry. At the same time, the absorbance at 600 nm was measured, and the content of MDA was calculated by the difference between the absorbance at 532 nm and 600 nm. Briefly, 0.1 g of the sample was weighed, added with 1 mL of extract solution, and homogenized in an ice bath. The extract solution was centrifuged at 4 °C, 8000× *g* for 10 min, and the supernatant was collected and placed on ice for the test.

Net photosynthesis was measured using Plant Photosynthesis Analyzer 3051D by irradiating at 400 μmol·m^−2^·s^−1^ and 25 °C (Top, Hangzhou, China). The machine was preheated until the carbon dioxide value of the measurement interface was stabilized. The measurement interval was 5 s, and the leaf chamber area was 11 cm^2^. The pulse-amplitude-modulated fluorometer (OS5p+ Modulated Chlorophyll Fluorometer, Aozuo Ecology Instrumentation Ltd., Beijing, China) was used to analyze the photochemical efficiency of PSII (ΦPSII) according to the method of Li et al. [59]. The chlorophyll fluorescence parameters were calculated by Yield = ΦPSII = ΔF/Fm’ = (Fm’ − F)/Fm’, where Fm’ is instantaneous maximum fluorescence observed under the saturation pulse radiation and F is the spontaneous fluorescence of light-adapted samples.

### 4.6. Statistical Analysis

Eight independent biological replicates were performed for lipidomic analysis and three independent biological replicates for physiological measures. The results were expressed as mean ± standard error (SE). SPSS 17 (IBM, New York, NY, USA) was used for one-way analysis of variance to test significant differences. The difference was considered statistically significant if *p* < 0.05.

## 5. Conclusions

In this study, our findings indicated that the changes of lipid molecular species associated with photosynthesis, membrane lipid peroxidation, and degradation, especially DG (29:1), SQDG (17:5/23:2), SQMG (16:0), MGMG (16:0), and MGDG (16:1/18:3), may act as biomarkers in response to drought tolerance in *N. flagelliforme.* The integrity of the photosynthetic membrane structure was maintained by membrane lipid remodeling and polyunsaturated fatty acid accumulation under drought stress. In addition, the elevated DGDG/MGDG ratio was promoted to maintain the stability of the thylakoid membrane under drought stress. Co (Q10) mitigates lipid peroxidation caused by ROS. Lipid degradation actively responds to drought stress by producing large amounts of energy and rationally allocating substrates. The increase in the OCFA level was also a key strategy in *N. flagelliforme* response to drought stress. The membrane lipid degradation and remodeling were possible mechanisms that led to drought tolerance in *N. flagelliforme*. Finally, the resistance biomarkers identified in *N. flagelliforme* could be used as indicators of drought stress response for cyanobacteria, and also could help us to achieve more insight into drought tolerant mechanism in this genus. At the same time, our results have proved that the untargeted lipidomics approach can serve as an important comprehensive tool to investigate the distribution and dynamic changes of OCFAs widely applied in agriculture and food.

## Figures and Tables

**Figure 1 foods-11-01798-f001:**
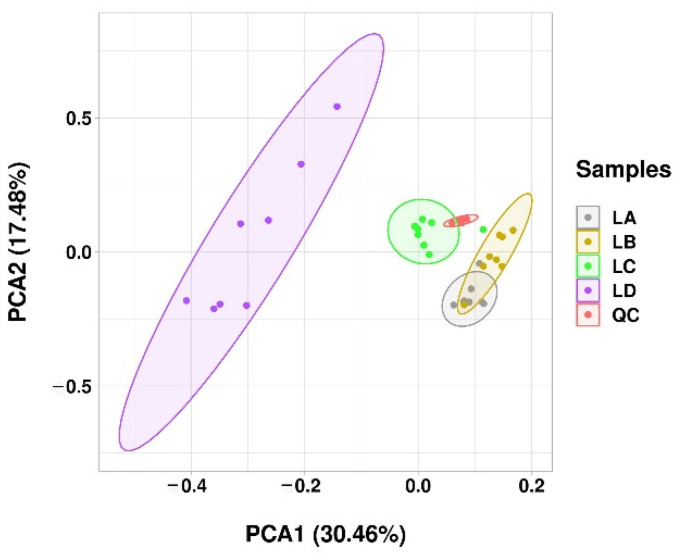
PCA analysis of lipids in *N. flagelliforme* under various water loss points. LA, LB, LC, and LD represent samples with 0%, 30%, 75%, and 100% water loss, respectively. The QC samples were used to monitor and evaluate the stability and reliability of the experimental data. The dot in the ellipse represents independent biological replicates.

**Figure 2 foods-11-01798-f002:**
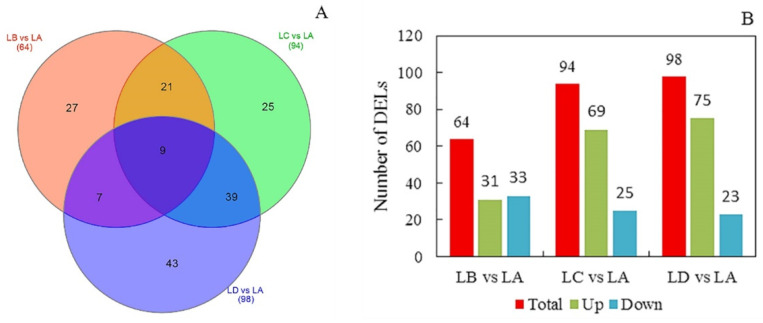
Distribution of identified DELs of *N. flagelliforme* in different stress groups compared to that in the control group. (**A**). Venn diagram showing the shared or unique DELs in stress groups vs. control group profiles. (**B**). Statistical analysis of DELs from three pairwise comparisons ((LB vs. LA, LC vs. LA, and LD vs. LA). LA, LB, LC, and LD represent samples with 0%, 30%, 75%, and 100% water loss, respectively. The total represents the total number of DELs identified in the comparison group, and the up and down represent the number of DELs up-regulated and down-regulated in the treatment group compared to the control group, respectively.

**Figure 3 foods-11-01798-f003:**
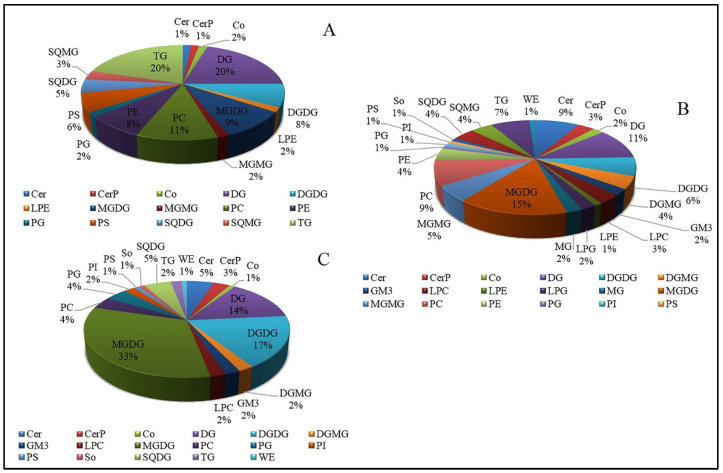
The classification of DELs in *N. flagelliforme* under drought stress. (**A**). The proportions of lipid species with significant differences in the LB group in comparison with the LA group. (**B**). The proportions of lipid species with significant differences in the LC group in comparison with the LA group. (**C**). The proportions of lipid species with significant differences in the LD group in comparison with the LA group. Lipid classes are presented as the percentage based on the total amount of DELs. LA, LB, LC, and LD represent samples with 0%, 30%, 75%, and 100% water loss, respectively.

**Figure 4 foods-11-01798-f004:**
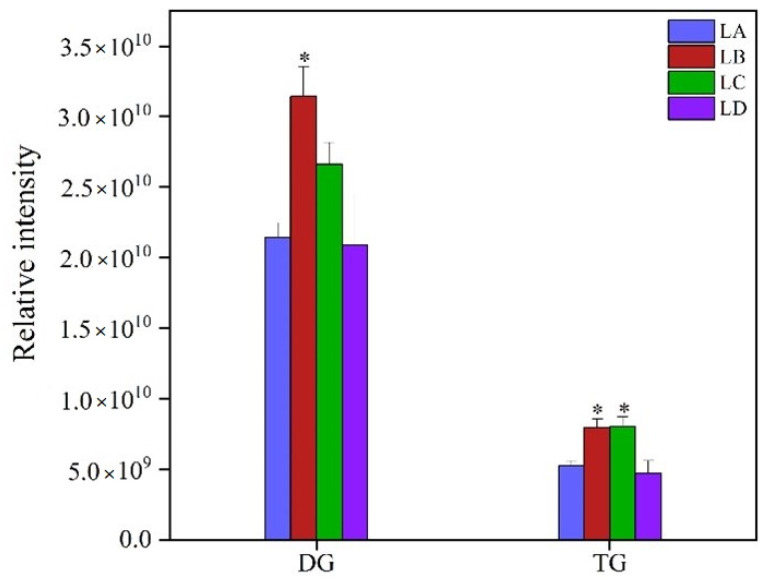
Changes of glycerolipid contents in *N. flagelliforme* under drought stress. LA, LB, LC, and LD represent samples with 0%, 30%, 75%, and 100% water loss, respectively. Values were presented as the means ± SE of eight independent biological samples. According to Student’s *t*-test (* *p* < 0.05), asterisks above the bars indicate significant differences between the treatment groups (LB, LC, and LD) and the control group (LA).

**Figure 5 foods-11-01798-f005:**
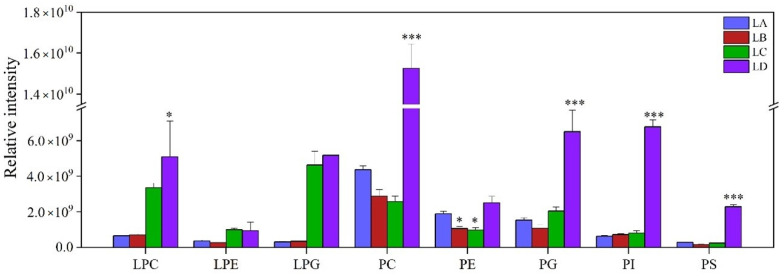
Changes of glycerolphospholipid contents in *N. flagelliforme* under drought stress. LA, LB, LC, and LD represent samples with 0%, 30%, 75%, and 100% water loss, respectively. Values were presented as the means ± SE of eight independent biological samples. According to Student’s *t*-test (* *p* < 0.05 and *** *p* < 0.001), asterisks above the bars indicate significant differences between the treatment groups (LB, LC, and LD) and the control group (LA).

**Figure 6 foods-11-01798-f006:**
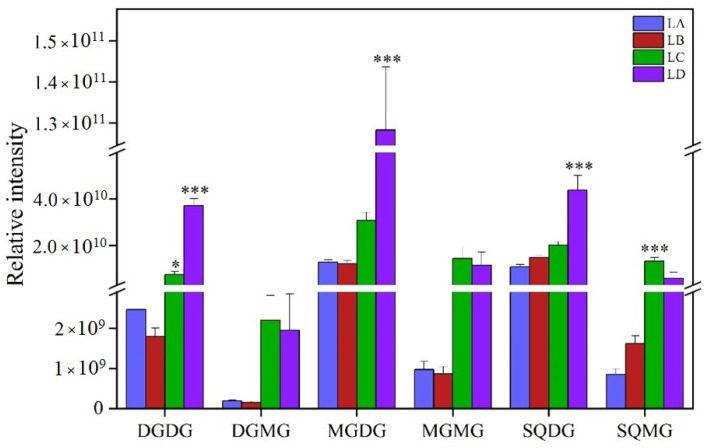
Changes of saccharolipid contents in *N. flagelliforme* under drought stress. LA, LB, LC, and LD represent samples with 0%, 30%, 75%, and 100% water loss, respectively. Values were presented as the means ± SE of eight independent biological samples. According to Student’s *t*-test (* *p* < 0.05 and *** *p* < 0.001), asterisks above the bars indicate significant differences between the treatment groups (LB, LC, and LD) and the control group (LA).

**Figure 7 foods-11-01798-f007:**
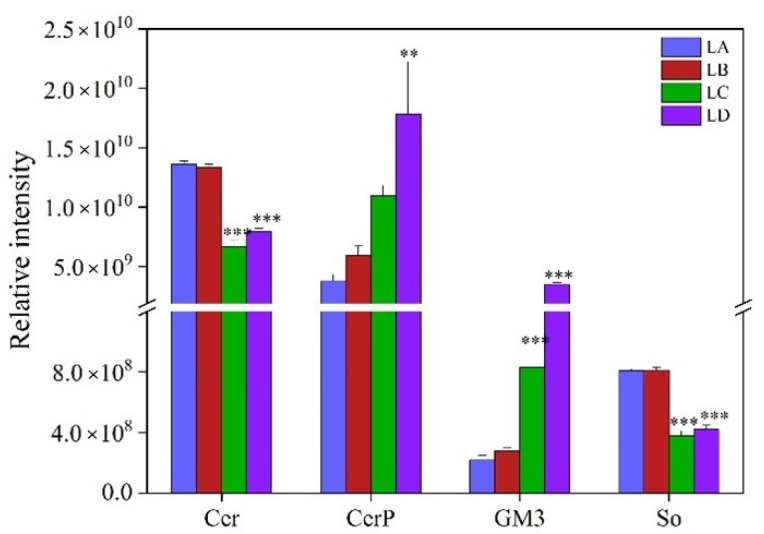
Changes of sphingolipid contents in *N. flagelliforme* under drought stress. LA, LB, LC, and LD represent samples with 0%, 30%, 75%, and 100% water loss, respectively. Values were presented as the means ± SE of eight independent biological samples. According to Student’s *t*-test (** *p* < 0.01 and *** *p* < 0.001), asterisks above the bars indicate significant differences between the treatment groups (LB, LC, and LD) and the control group (LA).

**Figure 8 foods-11-01798-f008:**
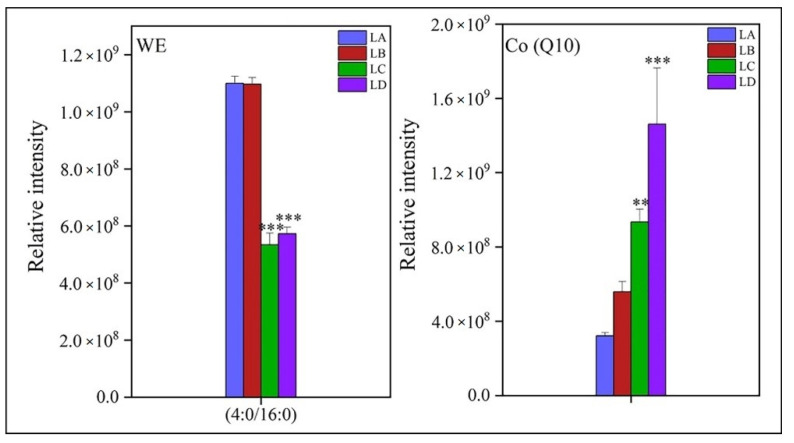
Changes of fatty acyl and prenol lipid contents and fatty acid compositions in *N. flagelliforme* under drought stress. LA, LB, LC, and LD represent samples with 0%, 30%, 75%, and 100% water loss, respectively. Values were presented as the means ± SE of eight independent biological samples. According to Student’s *t*-test (** *p* < 0.01 and *** *p* < 0.001), asterisks above the bars indicate significant differences between the treatment groups (LB, LC, and LD) and the control group (LA).

**Figure 9 foods-11-01798-f009:**
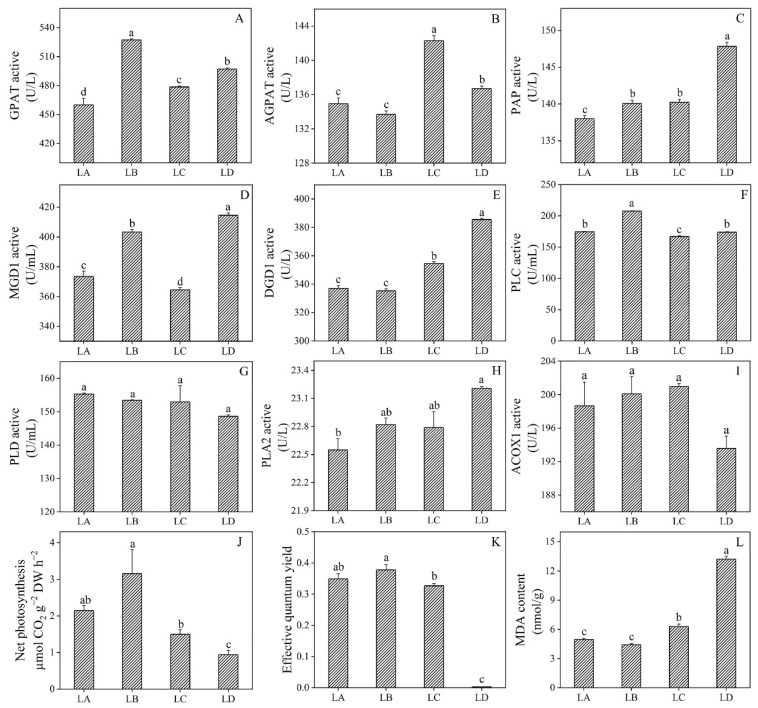
Physiological response of *N. flagelliforme* to persistent drought stress. Subfigures (**A**–**I**) show the activity of enzymes related to lipid metabolism in *N. flagelliforme* under drought stress. In addition, subfigures (**J**,**K**) represent net photosynthesis and photosystem II fluorescence parameters of *N. flagelliforme* during drought stress, respectively. Subfigure (**L**) displays the MDA content in *N. flagelliforme* during drought stress. LA, LB, LC, and LD represent samples with 0%, 30%, 75%, and 100% water loss, respectively. Values were presented as the means ± SE (*n* = 3). The different letters above the bars indicate that the means were significantly different (*p* < 0.05).

**Figure 10 foods-11-01798-f010:**
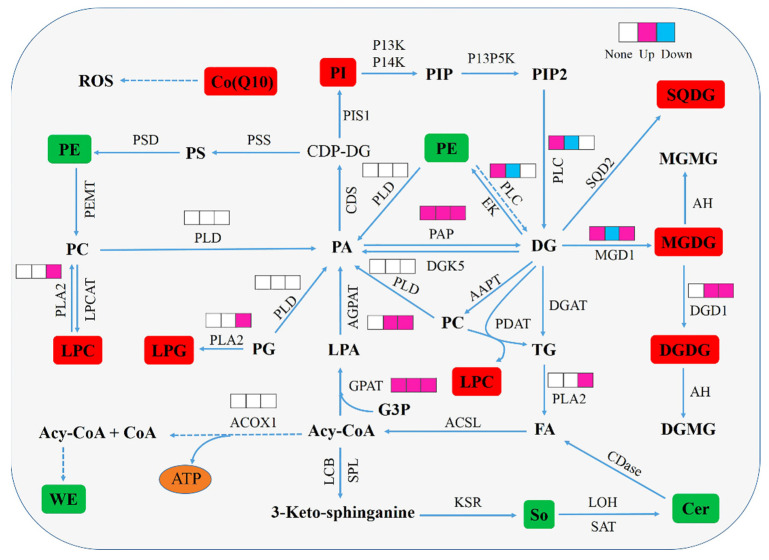
A model illustrating potential regulation mechanism of lipid metabolism in *N. flagelliforme* under drought stress. The red boxes represent the up-regulated metabolites, and the green boxes represent the down-regulated metabolites. The small squares arranged from left to right indicate the enzyme activity changes in LB, LC, and LD groups compared with those of the control group, respectively. White indicates no significant difference, pink indicates up-regulation, and blue indicates down-regulation (*p* < 0.05). Solid arrows indicate single process and dashed ones indicate multiple process.

## Data Availability

Data not presented in the manuscript or Appendix A can be provided if available.

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
