# Peer review of "Effect of Drought Stress on Degradation and Remodeling of Membrane Lipids in Nostoc flagelliforme"

_foods, 2022, doi:10.3390/foods11121798_

Round 1
Reviewer 1 Report
The study of lipidoma in various cultures under stress is a way to decipher the important mechanisms of cell resistance to the action of various degrading factors. The structural and functional integrity of cells is largely determined by the integrity of biological membranes. In this regard, the study of changes in the molecular footprint of lipidoma of one of the species of drought-resistant cyanobacteria is of great importance for revealing the general mechanisms of resistance to this phenomenon.
The article reflects an extensive analysis of the qualitative and quantitative composition of N. flagelliforme lipids under conditions of insufficient moisture at various levels. The results obtained allow the identification of resistance markers, as well as general mechanisms to ensure the survival of cyanobacteria in drought conditions.
Recommendations for improvement:
1. There are many abbreviations in the article, some of them are general, and some are private. Their abundance in the text of the article makes it difficult to trace the main idea. In order to make it easier to read and follow the abbreviations, I recommend listing them at the beginning of the article (those in table s6 and others).
2. The introduction is too voluminous. Much of the material in this compartment can be moved to the discussion section.
3. Conclusions should end with recommendations, for example, on the use of identified resistance markers.
Author Response
Response to the comments of reviewers
Dear reviewers,
Thank you for your letter and comments concerning our manuscript entitled “Effect of Drought Stress on Degradation and Remodeling of Membrane Lipids in Nostoc flagelliforme” (ID: foods-1748471). All the comments are valuable and very helpful for revising and improving our manuscript. We have revised the manuscript according to the your comments. Please find our responses as follows:
- There are many abbreviations in the article, some of them are general, and some are private. Their abundance in the text of the article makes it difficult to trace the main idea. In order to make it easier to read and follow the abbreviations, I recommend listing them at the beginning of the article (those in table s6 and others).
Respond: Thank you very much for your suggestion. We have compiled all the abbreviations in the article (including Table S6) and listed them before the Introduction.
- The introduction is too voluminous. Much of the material in this compartment can be moved to the discussion section.
Respond: We have revised the Introduction and moved some contents to the Discussion.
- Conclusions should end with recommendations, for example, on the use of identified resistance markers.
Respond: Thank you very much for your suggestion. We have added a recommendation for resistance markers at the end of the Conclusion.
We appreciate the reviewers’ comments that made this a better paper. Looking forward to hearing from you.
Kind Regards,
Dr. Wenyu Liang
School of Life Sciences
Ningxia University
Yinchuan 750001, P.R. China
E-mail Address: liang_wy@nxu.edu.cn
Reviewer 2 Report
1. The text contains multiple abbreviations. They are necessary, along with an initial definition list that would explain each acronym.
2. Although very informative, the illustrations and the figures are barely visible, even in the online format. If possible, publishing a better version of the illustrations will facilitate the analysis of the results.
3. The principle of the method is required in the case of an announcement by the kit manufacturer. The text talks about the preparation of biological matter for analysis.
4. Considering the volume of the received results, the conclusions are insufficient. Some recommendations for the results applications are needed. For example, a new way was obtained by modeling lipid content in cyanobacterial biomass.
Author Response
Response to the comments of reviewers
Dear reviewers,
Thank you for your letter and comments concerning our manuscript entitled “Effect of Drought Stress on Degradation and Remodeling of Membrane Lipids in Nostoc flagelliforme” (ID: foods-1748471). All the comments are valuable and very helpful for revising and improving our manuscript. We have revised the manuscript according to the your comments. Please find our responses as follows:
Comments and Suggestions for Authors
- The text contains multiple abbreviations. They are necessary, along with an initial definition list that would explain each acronym.
Respond: Thank you very much for your suggestion. We have compiled all the abbreviations in the article (including Table S6) and listed them before the Introduction.
- Although very informative, the illustrations and the figures are barely visible, even in the online format. If possible, publishing a better version of the illustrations will facilitate the analysis of the results.
Respond: We have rearranged the poorly observed figures in the text as much as possible (as shown in Figure 4 to Figure 7) and ensured sufficient resolution.
- The principle of the method is required in the case of an announcement by the kit manufacturer. The text talks about the preparation of biological matter for analysis.
Respond: We have added the principle of the method about enzyme activity and malondialdehyde content detection in the Materials.
- Considering the volume of the received results, the conclusions are insufficient. Some recommendations for the results applications are needed. For example, a new way was obtained by modeling lipid content in cyanobacterial biomass.
Respond: Thank you very much for your suggestion. We have added the recommendations for the results applications in Conclusion, as you suggested.
We appreciate the reviewers’ comments that made this a better paper. Looking forward to hearing from you.
Kind Regards,
Dr. Wenyu Liang
School of Life Sciences
Ningxia University
Yinchuan 750001, P.R. China
E-mail Address: liang_wy@nxu.edu.cn
Reviewer 3 Report
This study is devoted to the analyzis of the lipidome changes of cyanobacteria Nostoc flagelliforme under dehydration. By using the ultra-high-performance liquid chromatography and
mass spectrometry the authors have identified a total of 853 lipid molecules. The Authors performed an analysis of 171 lipid molecules that were significantly different from that of the control group. The Authors offered a lipid metabolic model that demonstrates the regulatory mechanism of drought stress in Nostoc flagelliforme. The study was conducted competently, the methods are described in sufficient detail. The figures and diagrams represent the results well.
Several comments are intended to help improve the text of the manuscript.
1) Introduction.
We can advise the Authors to move part of the text from the Introduction to the Discussion. The Introduction is too long.
At the end of the Introduction, the purpose of the work is usually formulated and the conclusion from the results obtained is made in the Conclusions section.
2) Materials and Methods.
Please indicate the geographical coordinates of the sampling site.
Please indicate whether the growth medium BG11 contained nitrogen or was nitrogen-free.
Did the filaments contain cyanobacteria heterocysts or were all the cells vegetative?
What are the statistics of the conducted studies? How many independent biological repeats have been done? Please indicate.
Author Response
Response to the comments of reviewers
Dear reviewers,
Thank you for your letter and comments concerning our manuscript entitled “Effect of Drought Stress on Degradation and Remodeling of Membrane Lipids in Nostoc flagelliforme” (ID: foods-1748471). All the comments are valuable and very helpful for revising and improving our manuscript. We have revised the manuscript according to the your comments. Please find our responses as follows:
Several comments are intended to help improve the text of the manuscript.
1) Introduction.
We can advise the Authors to move part of the text from the Introduction to the Discussion. The Introduction is too long.
At the end of the Introduction, the purpose of the work is usually formulated and the conclusion from the results obtained is made in the Conclusions section.
Respond: Thank you very much for your suggestion. We have revised the Introduction and moved some contents to the Discussion. In addition, we clearly pointed out the purpose of this work in the end of the Introduction, and summarized the results in the Conclusion section.
2) Materials and Methods.
Please indicate the geographical coordinates of the sampling site. Please indicate whether the growth medium BG11 contained nitrogen or was nitrogen-free. Did the filaments contain cyanobacteria heterocysts or were all the cells vegetative? What are the statistics of the conducted studies? How many independent biological repeats have been done? Please indicate.
Respond: We added the geographic coordinates of the sampling site in the Materials. The BG11 medium contained nitrogen (containing 1.5 g·L−1 NaNO3), and we have supplemented this point in the text. All experiments were carried out with intact algal filaments, including heteromorphic and vegetative cells. Furthermore, eight independent biological replicates of N. flagelliforme were used for lipidomics, and three replicates for physiological index measurement, including enzyme activity, MDA content and photosynthetic parameters. SPSS 17 software was used to analyze the significance of lipid molecular content and physiological indexes. We have indicated this point in the text.
We appreciate the reviewers’ comments that made this a better paper. Looking forward to hearing from you.
Kind Regards,
Dr. Wenyu Liang
School of Life Sciences
Ningxia University
Yinchuan 750001, P.R. China
E-mail Address: liang_wy@nxu.edu.cn